# Holocene sea ice variability driven by wind and polynya efficiency in the Ross Sea

K. Mezgec[1,2], B. Stenni [3,4], X. Crosta[5], V. Masson-Delmotte[6], C. Baroni[7,8], M. Braida[2], V. Ciardini[9], E. Colizza[2], R. Melis[2], M.C. Salvatore[7,8], M. Severi[10], C. Scarchilli[9], R. Traversi [10], R. Udisti[10] & M. Frezzotti [9]

The causes of the recent increase in Antarctic sea ice extent, characterised by large regional contrasts and decadal variations, remain unclear. In the Ross Sea, where such a sea ice increase is reported, 50% of the sea ice is produced within wind-sustained latent-heat polynyas. Combining information from marine diatom records and sea salt sodium and water isotope ice core records, we here document contrasting patterns in sea ice variations between coastal and open sea areas in Western Ross Sea over the current interglacial period. Since about 3600 years before present, an increase in the efficiency of regional latent-heat polynyas resulted in more coastal sea ice, while sea ice extent decreased overall. These past changes coincide with remarkable optima or minima in the abundances of penguins, silverfish and seal remains, confirming the high sensitivity of marine ecosystems to environmental and especially coastal sea ice conditions.

[1] Department of Physical Sciences, Earth and Environment, University of Siena, via del Laterino 8, 53100 Siena, Italy. [2] Department of Mathematics and Geosciences, University of Trieste, via E. Weiss 2, 34128 Trieste, Italy. [3] Department of Environmental Sciences, Informatics e Statistics, Ca' Foscari University of Venice, via Torino 155, 30172 Venezia, Italy. [4] Institute for the Dynamics of Environmental Processes—CNR, via Torino 155, 30172 Venezia, Italy. [5] UMR-CNRS 5805 EPOC, Université de Bordeaux, Allée Geoffroy Saint Hilaire, 33615 Pessac cédex, France. [6] LSCE (IPSL, CEA-CNRS-UVSQ, Université Paris Saclay), Bat 701 L'Orme des Merisiers, CEA Saclay, 91191 Gif-sur-Yvette cédex, France. [7] Department of Earth Sciences, University of Pisa, Via S. Maria 53, 56126 Pisa, Italy. [8] Institute of Geosciences and Earth Resources—CNR, Via G. Moruzzi 1, 56124 Pisa, Italy. [9] ENEA, SP Anguillarese 301, 00123 Roma, Italy. [10] Department of Chemistry "Ugo Schiff", University of Florence, Sesto Fiorentino, 50019 Firenze, Italy. Correspondence and requests for materials should be addressed to B.S. (email: barbara.stenni@unive.it)

Antarctic sea ice is an important component of the climate system and its seasonal cycle affects global climate dynamics because of its interplay with the planetary albedo, atmospheric circulation, ocean productivity and thermohaline circulation through Antarctic bottom water (AABW) formation[1].

Direct knowledge of Antarctic sea ice variability is restricted to the satellite era of the past 39 years (November 1978)[2] and limited historical information[3]. During this era, sea ice cover and duration have increased in the Ross Sea (RS) region, which is one of the three main sources of AABW, in contrast to opposite trends in the adjacent Amundsen–Bellinghausen Sea[2]. Simulations performed with state-of-the-art climate models fail to capture such a sea ice increase[4]. The mechanisms that drive Antarctic sea ice changes, and the respective roles of internal climate variability vs. natural and anthropogenic forcing remain debatable[5, 6]. Moreover, polar marine ecosystems appear particularly sensitive to changes in the extent and thickness of sea ice[7], although much remains to be understood about their past and present responses under rapid environmental changes.

Each year, the RS exports by geostrophic winds and ocean currents more than twice its own surface area of pack ice during the average 9-month ice growing period. More than 50% of this pack ice production occurs within the RS and Terra Nova Bay (TNB) latent-heat polynyas[8] (Fig. 1), where newly formed sea ice is pushed to the leeward side by persistently strong katabatic winds while the coastal windward side remains ice-free. Today, mainly two types of sea ice are observed in the Western Ross Sea (WRS). The first, is pack ice (<1 m depth), which is formed in the WRS and is transported northward by geostrophic winds and currents. The second, is coastal sea ice (landfast ice and persistent pack ice), which is attached to the shoreline or anchored by icebergs[9], it is generally thicker (>1.0 m depth) than pack ice due to the accretion of platelet ice underneath the congelation ice[10]. Platelet ice formation is strictly related to the efficiency of the coastal polynyas[11] to produce sea ice and ice shelf water (ISW), and is mainly driven by katabatic wind persistence. At the seasonal and inter-annual scales, no relationship has been observed between coastal ice and pack ice extensions in the WRS[9].

Landfast ice usually forms in early autumn (March–April) and melts back in late summer (January) when the protective surrounding pack ice has waned. A delay of about 45 days is observed between minimum pack ice extent and landfast ice melting/breaking. The annual landfast ice has a thickness of up to 2.5 m[12, 13] and is formed by columnar and granular ice plus consolidated platelet ice[10, 14]. Consolidated platelet ice can form more than 50% of the landfast ice column[10, 15] by accreting underneath. The accretion of platelets below congelation ice starts during the mid-austral winter due to the advection of supercooled ISW from underneath floating glaciers and shelf cavities. This process provides an oceanic heat sink for persistent growth of frazil crystals as a first stage for platelet ice formation[14, 16]. Supercooled ISW has been widely observed along the WRS coast[16, 17], associated to thermohaline circulation below ice shelves due to the interaction of high-salinity shelf water (HSSW) melting the ice at the base of the ice shelves. This HSSW mainly forms in the latent-heat polynyas of TNB and RS due to high salt fluxes into the ocean associated with continuous sea ice formation. Only episodically landfast ice can persist for several years along Victoria Land coast (VLC)[18]. On the other hand, the pack ice is characterised by a thickness of <1 m and is mainly constituted of columnar and granular ice[19]. This structure is related

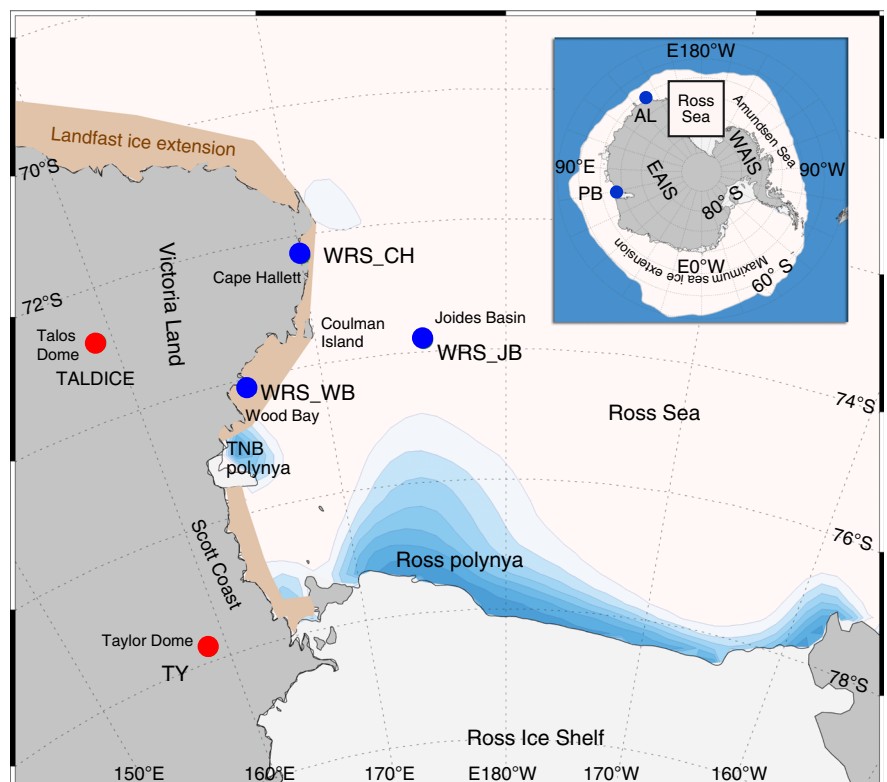

**Fig. 1** Location of the studied marine and ice core sites in the WRS area. The TALDICE and Taylor Dome (TY) ice cores and the Joides Basin (WRS_JB), Cape Hallett (WRS_CH) and Wood Bay (WRS_WB) sediment cores are indicated in the map. The Ross Sea and Terra Nova Bay (TNB) latent-heat polynyas (from cyan to blue, % of open water from May to November) and present day landfast ice extension area (brown) are schematically represented. The inset shows a map of Antarctica with the location of the other marine cores cited in the text and retrieved in the Adélie Land (AL) and Prydz Bay (PB)[38]. East Antarctica ice sheet (EAIS), West Antarctica ice sheet (WAIS) and the maximum Antarctic winter sea ice extent is depicted

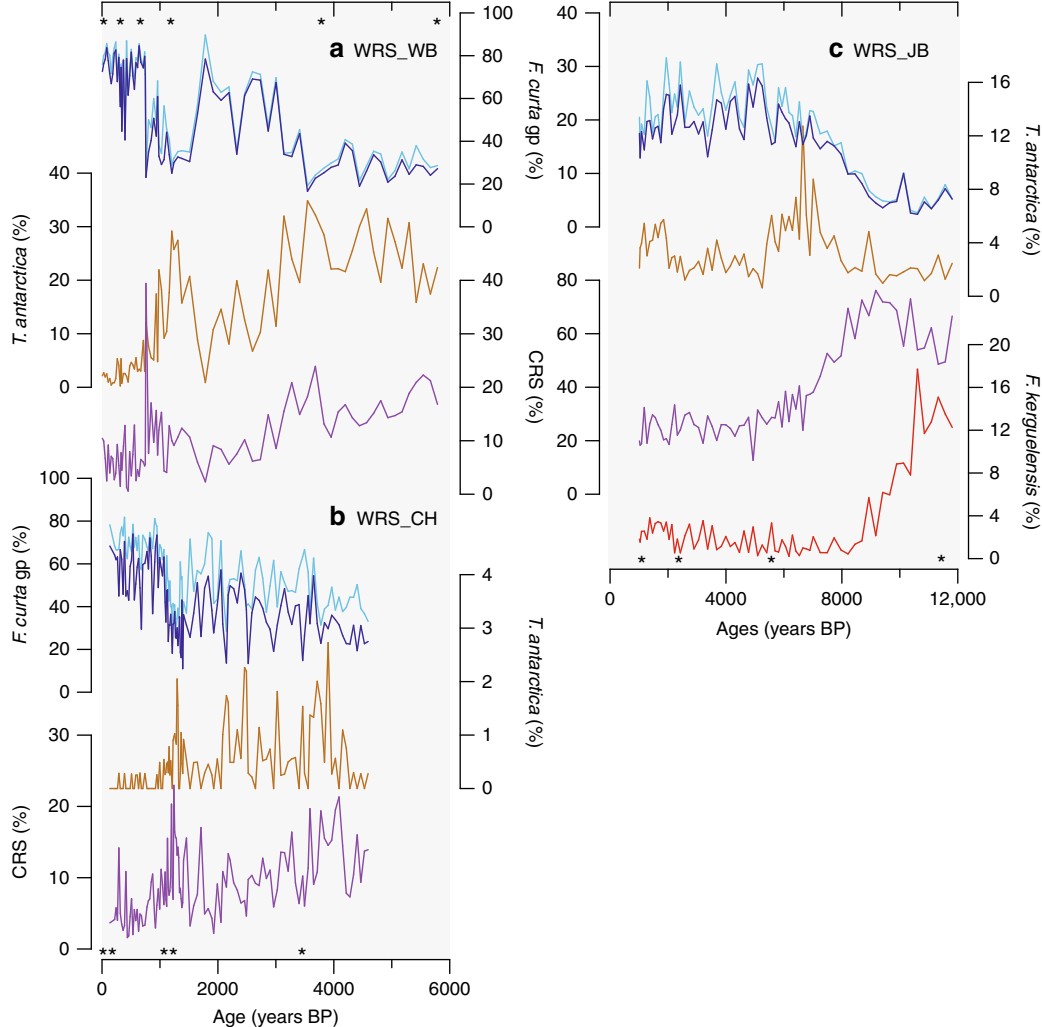

**Fig. 2** Holocene diatom records. **a** Diatom relative abundances of *Fragilariopsis curta* (dark blue line) and the *F. curta* + *F. cylindrus* gp (light blue line), *Thalassiosira antarctica* resting spores (brown line) and *Chaetoceros Hyalochaete* resting spores (purple line) in core WRS_WB (BAY05-43c) from Wood Bay. **b** Similar to **a** but in core WRS_CH (BAY05-20c) from Cape Hallett. **c** Similar to **a** but in core WRS_JB (ANTA99-cJ5) from Joides Basin, with the addition of *F. kerguelensis* relative abundances (red line). Stars represent the calibrated [14]C ages for each marine core

to the accretion of congelation ice from seawater with ample evidence of a thermodynamic process driven mainly by air and ocean temperatures[14].

This study aims to better understand the main factors controlling WRS sea ice types and dynamic changes during the past millennia on a centennial timescale using three marine core records and two nearby East Antarctic ice core records (Fig. 1) as a benchmark to recent sea ice increase in WRS[2]. Reconstruction of past sea ice is based on diatom analyses in marine cores and sea salt sodium (ssNa) in ice cores, while Antarctic climate is inferred from ice core water isotope records. Three gravity cores ANTA99-cJ5, BAY05-20c and BAY05-43c (hereafter named WRS_JB, WRS_CH and WRS_WB, respectively) were collected from the WRS continental shelf. Two cores were collected in the coastal sea ice zone (CSIZ) at Cape Hallett (WRS_CH) and Wood Bay (WRS_WB), in an area characterised by annual landfast ice[18], and one was retrieved in the open sea ice zone (OSIZ) at Joides Basin (WRS_JB), where pack ice generally melts and breaks up in early December.

The current study documents contrasting patterns between open and coastal sea ice during the Mid–Late Holocene in the WRS, and an increase in regional polynyas' efficiency starting from ~3600 years before present (1950 AD, hereafter ka). These

Holocene changes in pack ice cover, polynya and coastal sea ice are expected to have affected the presence of seals and penguins, animals known to be sensitive to changes in coastal sea ice conditions[7]. To show this, we compared our new reconstructions to changes in the abundance of three sea ice-dependent fauna (Adélie penguins, elephant seals and silverfish) as documented by radiocarbon-dated coastal remains along VLC[20, 21].

## Results

**The marine archives**. The age models of the marine cores are based on calibrated [14]C dates performed on acid insoluble organic matter. Overall, the chronology of the three marine cores is robust to within ~600 years, after accounting for uncertainties in the raw dates, regional marine reservoir ages plus local dead carbon (DC) contamination corrections as well as down-core extrapolation. Moreover, the comparisons between ice and marine records have been performed using a 200-year data resampling, that reduced drastically the uncertainty in the correlations. Information on the dating technique, reservoir ages and DC corrections along with the accuracy of the chronologies is detailed in the Methods and Supplementary Note 1. Cores WRS_CH and WRS_WB cover the last 4.6 and 5.8 ka, respectively, while

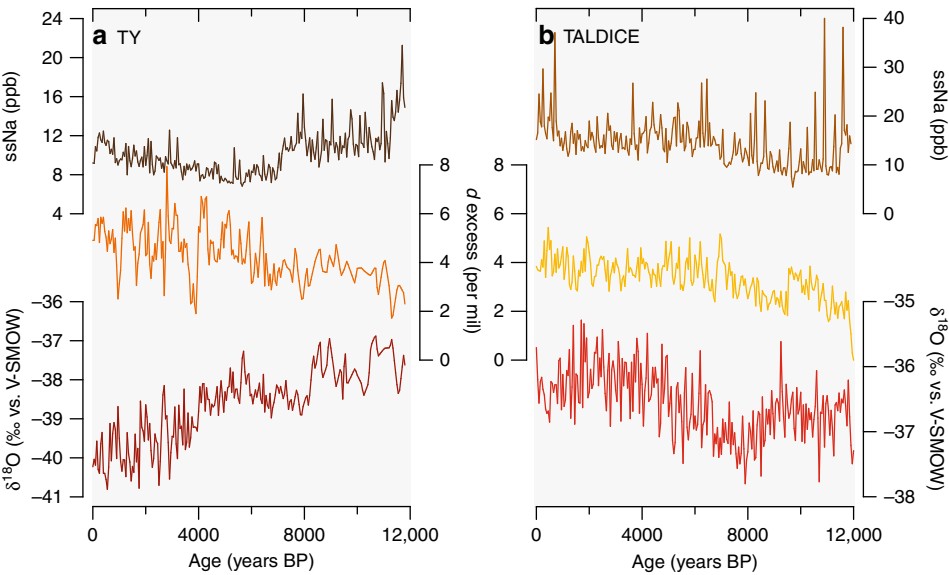

**Fig. 3** Holocene ice core records. **a** sea salt sodium concentration in part per billion (dark brown line), deuterium excess in ‰ vs. V-SMOW (orange line) and δ[18]O in ‰ vs. V-SMOW (dark red line) in Taylor Dome ice core[33–35]. **b** sea salt sodium concentration in part per billion (brown line), deuterium excess in ‰ vs. V-SMOW (yellow line) and δ[18]O in ‰ vs. V-SMOW (red line) in TALDICE ice core. All ice core records were resampled on a 50-year time step

WRS_JB covers the last 11.8 ka (Fig. 2). Diatom counts are performed with a mean resolution of 40, 70 and 150 years, respectively.

In the Southern Ocean, qualitative reconstructions of sea ice duration are generally achieved using down-core relative abundances of the *Fragilariopsis curta* group (*F. curta* and *Fragilariopsis cylindrus*)[24]. Both diatoms are sea ice-related species reaching highest abundances both in plankton population and surface sediments in locations that experience 9–11 months of sea ice cover and highly consolidated winter sea ice conditions[22–24]. Here we use solely *F. curta* relative abundances (Fig. 2) due to the predominance of this species (mean value 83%) and to remain consistent with regional studies[23, 25]. Other diatom species are used to refine environmental reconstructions inferred from *F. curta* records (Fig. 2). *Fragilariopsis kerguelensis* is an open ocean diatom reaching highest abundances in plankton population and surface sediments around the polar front zone[26]. Abundances decrease southward with increasing sea ice duration and surface water cooling. In Antarctic coastal regions, *Chaetoceros Hyalochaete* forms resting spores (CRS) during events of productivity-triggered nutrient depletion in highly stratified surface waters due to ice melting[24, 27]. At very high southern latitudes, *Thalassiosira antarctica* thrives during the summer and forms resting spores (TRS) when sea ice returns in autumn[24], therefore tracking long/warm summers followed by cold autumns[28]. *F. curta* relative abundances in the two coastal records, WRS_WB and WRS_CH, increase from ~20% before 3.6 ka to ~70–80% over the last 1000 years (Fig. 2a, b). Between ~5.8 and ~3.6 ka the lowest *F. curta* relative abundances are concomitant to high occurrences of CRS and TRS. The subsequent increase in *F. curta* relative abundances, along with the decrease in CRS and TRS occurrences, is punctuated by a multi-centennial drop in *F. curta* relative abundances and a concomitant increase in CRS and TRS occurrences between 1.5 and 0.9 ka. Highest *F. curta* relative abundances are recorded over the last centuries.

In the WRS_JB core, *F. curta* relative abundances vary between ~10 and ~30% over the Holocene (Fig. 2c) and present an opposite pattern to *F. curta* records in the coastal cores. The lowest *F. curta* abundances are found between 11.8 and 8.2 ka,

concomitant with high occurrences of *F. kerguelensis* and CRS. *Fragilariopsis curta* relative abundances increase between 8.2 and ~5 ka while a congruent decrease in *F. kerguelensis* and CRS is observed. Subsequently, *F. curta* relative abundances decline back from ~30 to 15% at the core top, except for a small rebound centred around ~2 ka. This decline is accompanied by a slight increase of *F. kerguelensis* and TRS, while CRS presence stays very low.

The diatom records show sea ice dynamics in the coastal and open regions of WRS over the last millennia were different. Heavier sea ice conditions, indicating more persistent landfast ice and greater polynya efficiency, are inferred from 3.6 ka in the CSIZ concomitant with a reduction in pack ice duration and extent from 5 ka in the OSIZ.

**The glacial archives.** Fingerprints of WRS sea ice and Holocene climate changes can be provided by the nearby Talos Dome (TALDICE) and Taylor Dome (TY) (Fig. 1) ssNa and water isotope (δ[18]O and deuterium excess: $d = \delta D - 8 \times \delta^{18}O$) ice core records (Fig. 3). The production of new pack ice is the main source of ssNa aerosols through frost flower crystal formation and/or sublimation of salty blowing snow[29]. Stable water isotopes in ice core records are affected by changes in evaporation conditions and distillation that occur during air mass cooling towards Antarctica. The snowfall δ[18]O is a local temperature proxy related to climate and/or elevation changes with less/more negative values associated with higher/lower surface atmospheric temperatures[30]. The deuterium excess is a proxy of evaporative conditions (SST, relative humidity and wind speed) at moisture source regions[31] and can be considered as an integrated tracer of the hydrological cycle[30, 31]. A southern limit of sea ice would allow a significant contribution from high-latitude cold moisture sources with low *d* values. Conversely, a northern limit of sea ice would lead to the opposite effect. During the sea ice formation season, a southern shift of sea ice limit and/or an increase in size (efficiency) of latent-heat polynyas would decrease deuterium excess but increase δ[18]O values due to local (cooler) moisture availability. Conversely, an increase in new sea ice formation is expected to enhance the ssNa content in ice cores.

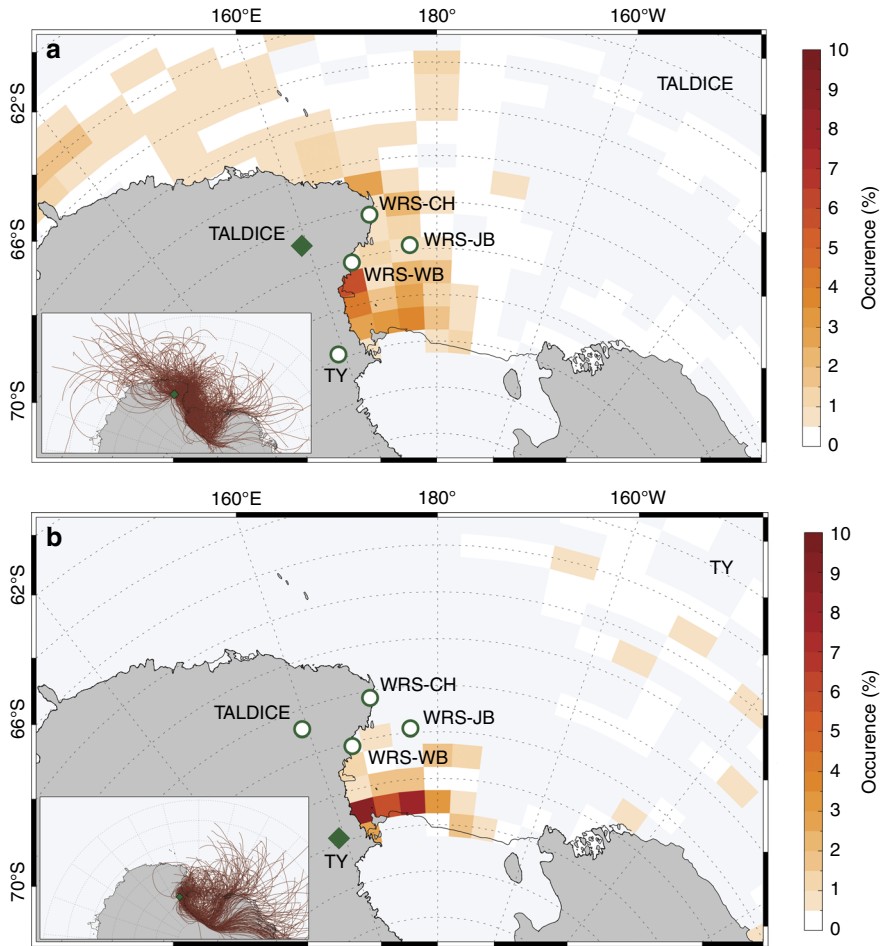

**Fig. 4** Sea salt aerosol source regions and paths from sea ice area toward ice core using model air mass trajectories. **a**, **b** occurrence (%) of loading condition for selected back trajectories. Back trajectories can be considered loaded above the sea ice if they flow above an area with sea ice values between 15 and 75% and wind speed >3 m s$^{-1}$. The occurrence of events is spatially cumulated for a regular grid of 5°×1° of longitude and latitude (the number of occurrences for each site is in percentage in respect to the total number of occurrences for the entire domain. The entire domain is defined as: 80°S<latitude<50°S; 0° <longitude<180°E−180°W. **a**, **b** Insets: 3 days back trajectories ending at 1000 m over TALDICE (**a** inset) and TY (**b** inset) and spending more than 10% of their path over the sea and <40% over the continent (red lines). Only trajectories related to March, April, May, June, July, August, September, October and November for the period 1979–2012 are considered. Among the selected trajectories, only those with a permanence of at least three consecutive hours over sea ice concentration>15% are considered. The green open diamond represents the location of TALDICE (**a**) and TY (**b**), whereas open circles indicate the location of the three marine cores (Cape Hallett WRS_CH; Wood Bay WRS_WB and Joides Basin WRS_JB) and the other ice core

To identify the ssNa and moisture sources in TALDICE and TY, we analysed the modern air mass trajectories that passed over the sea ice area before reaching those two ice core sites using atmospheric reanalyses and sea ice satellite data (Fig. 4; Methods). Due to the uncertainty associated to trajectories, it was possible to estimate the total error on the air mass path to the order of between 10 and 30% of the travel distance. For each ice core site, there is a low inter-annual variability of the selected back trajectories in terms of geographical origin and numbers. The calculated back trajectories only represent the dynamical main path of a hypothetical air volume moving in a three-dimensional wind field. This type of back trajectory analysis provides no information about chemical exchange or depositional phenomena. Hereafter, we assume that there is a persistent Holocene relationship between modern air mass trajectories and uplift and transport of species (water isotopes, sea salt aerosols) towards the ice core sites. The trajectories arriving at TY mainly transit above the WRS recurrent latent-heat polynyas. In contrast, the trajectories arriving at TALDICE (Fig. 4) sweep a wide new pack ice area of the RS and Southern West Pacific–East Indian Ocean[32].

Here we present published Holocene records of TY[33–35] (Fig. 3a) (mean Holocene resolution of 5 and 33 years for ssNa, $\delta^{18}$O and $d$, respectively) and new data from TALDICE (Fig. 3b) (mean resolution of 18 years for both ssNa, $\delta^{18}$O[36] and $d$; see Methods and Supplementary Figs. 4, 5 for raw data). The TY ssNa content is higher in the early Holocene and exhibits a two-step decrease until 6–7 ka, followed by a gradual increase until present (Fig. 3a). In contrast, the TALDICE ssNa is low at 10 ka, followed by an increase until 6.5 ka, with a subsequent multi-millennial plateau until 3.5 ka, a slight decrease between 3.5 and 1 ka, and a sharp increase to reach maximum values during the last millennium (Fig. 3b). Contrasted behaviours are also observed in the $\delta^{18}$O records, with TY decreasing throughout the Holocene and TALDICE exhibiting relative high values during the early Holocene followed by a minimum at ~8 ka, with an increasing trend from 8 to 1.5 ka and again lower values during the last millennium. Despite the differences in the $\delta^{18}$O trends, the $d$ records from both TY and TALDICE present similarly increasing trends over the Holocene (Fig. 3), though the TY $d$ record shows much greater variability since 7 ka (Supplementary Table 2). The TALDICE and TY $\delta^{18}$O records start to diverge at 7 ka,

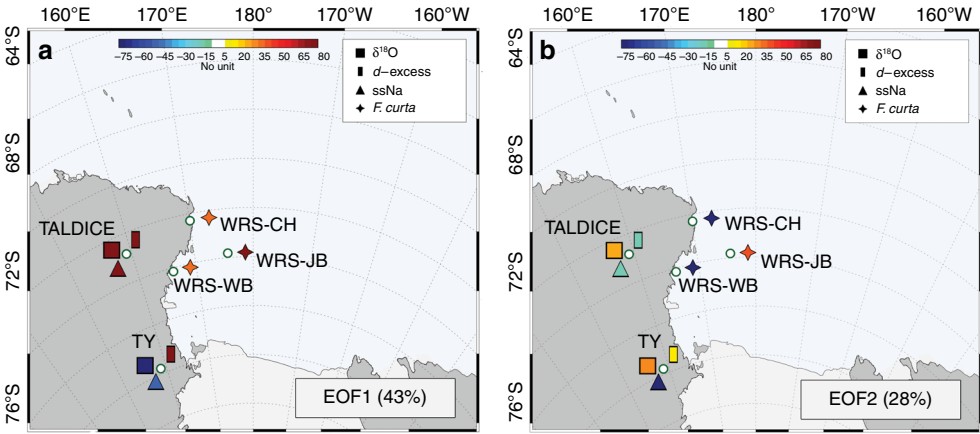

**Fig. 5** Empirical orthogonal function analysis of ice and marine records. Least square empirical orthogonal function analysis applied to a covariance matrix based on standardized values of δ18O (filled square), deuterium excess (d-excess, filled rectangle) and sea salt sodium (ssNa, filled triangle), obtained from TALDICE and Taylor Dome (TY) ice core data, and, *F. curta* (filled star) obtained from Cape Hallett (WRS_CH), Joides Basin (WRS_JB) and Wood Bay (WRS_WB) marine core data. The colour code shows the strength of the component values of EOF1 and EOF2 (**a** and **b**, respectively) for each proxy data used in the calculation. The numbers in brackets represent the percentages of variance explained by EOF1 and EOF2 (43% and 28% of the total variance, respectively)

concomitantly with the gradual increase of ssNa content in TY, stable ssNa in TALDICE and onset of large amplitude *d* variability at TY.

The contrasted behaviours of TY and TALDICE ssNa, and the water stable isotope records argue for a difference in the air mass source areas reaching the two sites. The ssNa and water isotope records show marked variability and difference after 7 ka linked to the opening of the RS and associate sea ice dynamics between coastal and open regions of WRS over the last millennia.

**Coherency between marine and ice core records**. An empirical orthogonal function analysis has been applied to the δ18O, *d*, ssNa and *F. curta* data (Methods) to elucidate the link between OSIZ and CSIZ sea ice proxy records from marine and glacial archives. The first component (EOF1) accounts for 43% of the common variance (Fig. 5a) with high positive correlation observed between the WRS_JB diatom record and the TALDICE, and TY *d* records ($r = 0.77$, $p < 0.001$, $r = 0.57$, $p < 0.001$, respectively) and the TALDICE ssNa record ($r = 0.49$, $p < 0.001$) (Fig. 5a; see Methods and Supplementary Table 3). EOF2 accounts for 28% of the common variance (Fig. 5b) and demonstrates a negative value for the WRS_CH and WRS_WB diatom records, and the TY ssNa record that are significantly correlated (TY ssNa vs. WRS_CH: $r = 0.67$, $p < 0.001$; TY ssNa vs. WRS_WB: $r = 0.71$, $p < 0.001$) and a high positive value for the WRS_JB diatom record (Fig. 5; see Methods and Supplementary Tables 4, 5).

## Discussion
The coherency between WRS_JB diatom, TALDICE and TY *d* and TALDICE ssNa Holocene records (Fig. 5a) suggests a contemporary increase in pack ice extent (ice core proxies) and duration (*F. curta* relative abundance) in the OSIZ area of WRS. Conversely, the negative strength of EOF2 (Fig. 5b), with significant correlations between WRS_CH and WRS_WB diatom and TY ssNa Holocene records, refers to the sea ice formation linked to latent-heat polynyas of the WRS. The EOF analysis supports the link between TY ssNa down-core record and Holocene changes in sea ice produced within the WRS latent-heat polynyas as well as TALDICE ssNa down-core record and Holocene changes in pack ice conditions in northern and central

WRS. Our results therefore suggest that diatoms from CSIZ and TY ssNa record are representative of new sea ice formation linked to RS and TNB polynya efficiency, while the OISZ diatoms and TD ssNa records are related to new pack ice formation in RS and Southern West Pacific–East Indian Ocean.

Both *F. curta* and ssNa records support the existence of different sea ice dynamics in the coastal and open regions of WRS over the last millennia (Fig. 6c–e). More precisely, low *F. curta* relative abundances in WRS_CH and WRS_WB along with low ssNa concentrations in TY between ~5.8 and ~3.6 ka suggest an early spring sea ice retreat and a long sea ice-free season related to loose platelet ice in coastal areas[26] in the proximity of TNB polynya (Fig. 6c, d). The proximity of WRS_WB to these platelet ice sources may explain higher *F. curta* relative abundances in this core than in WRS_CH[26] (Fig. 2a, b). As such, the low CSIZ *F. curta* relative abundances (<40%) and reduced TY ssNa concentrations could be associated with less efficient latent-heat polynyas (TNB and RS; Fig. 6b) and thinner and less persistent coastal sea ice. The *F. curta* and ssNa increase since ~3.6 ka indicates icier conditions in the CSIZ in phase with coastal glacier expansion and atmospheric cooling in East Antarctica coastal sites during the Neoglacial period[37]. This cold period is however interrupted by milder conditions with earlier spring sea ice melting and a longer growing season in the CSIZ during 1.5–0.9 ka period, as similarly reported in Adélie Land and Prydz Bay regions[38]. High CSIZ *F. curta* relative abundances (>60%) and high TY ssNa concentrations were associated with more efficient regional latent-heat polynyas and more pervasive landfast ice. We suggest that increases (decreases) in polynya efficiency, and related ISW formation under floating glaciers/ice shelves, are mostly due to stronger (weaker) katabatic winds and/or seaward position of Ross Ice Shelf (RIS) and Drygalski ice fronts.

Conversely, low relative abundances of *F. curta* in WRS_JB along with low concentrations of ssNa in TALDICE between 11.8 and 8.2 ka (Fig. 6e) reflect a perturbed seasonal sea ice cycle due to melting of the RIS and iceberg presence. Indeed, congruent high abundances of *F. kerguelensis* (Fig. 2c) suggest a strong intrusion of relatively warm circumpolar deep water[39] (CDW) that may have contributed to RIS recession[40]. This recession could have formed a calving bay[27] over the Joides Basin as suggested by the overwhelming dominance of CRS until 8 ka (Fig. 2c). As such, diatoms and ssNa records suggest pervasive sea

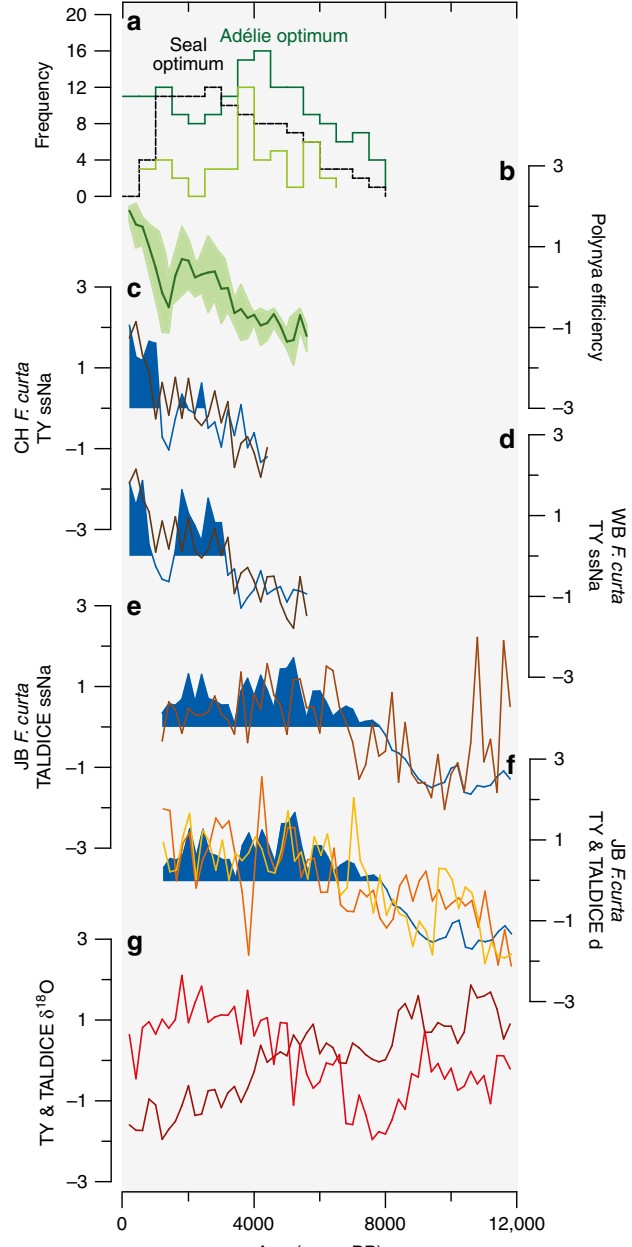

**Fig. 6** Holocene WRS sea ice variability in the CSIZ and OSIZ along with polynya efficiency and faunal remains. **a** Frequency of [14]C-dated faunal remains: Adélie penguins (dark green line), Antarctic silverfish (light green line) and elephant seals (black dashed line). **b** Polynya efficiency, with standard deviation values, obtained from a stacked record of CSIZ *F. curta* and TY ssNa records. **c** *Fragilariopsis curta* relative abundances in WRS_CH (blue line), with blue shading representing heavier CSIZ conditions, compared to TY ssNa[33] (dark brown line) as a coastal sea ice proxy. The correlation coefficient (*r*) between the two records is 0.67. **d** Similar to **c** but for WRS_WB with *r* equal to 0.71. **e** *F. curta* relative abundances in WRS_JB (blue line), with blue shading representing heavier, more extended OSIZ conditions, compared with TALDICE ssNa (light brown line) as a proxy for pack ice extent; *r* is 0.49. **f** *F. curta* relative abundances in WRS_JB (blue line and blue shading) compared with TY[35] (orange line) and TALDICE (yellow line) deuterium excess (*d*) records as proxies of evaporative conditions at moisture source regions and related to pack ice extent; *r* between JB and TY is 0.57 while *r* between JB and TALDICE is 0.77. **g** TY[34] (dark red line) and TALDICE (light red line) δ[18]O records as proxies of local site temperature variability. All data were resampled on a 200-yr time step and standardized for the considered periods

ice in spring but warm surface waters during short summers due to strong surface water stratification, intrusion of CDW and warm surface air[41] as already observed during the deglaciation of Prydz Bay[42]. After this period, increasing *F. curta* relative abundances and ssNa concentrations (Fig. 6e) indicate the establishment of a regular seasonal sea ice cycle and a greater sea ice extent as the RIS retreats away from the core site. Both proxy records suggest that larger sea ice extent in the OSIZ was reached by ~5 ka. The subsequent decrease in *F. curta* relative abundances and ssNa concentrations, associated with slightly increasing occurrences of *F. kerguelensis* and TRS (Fig. 2c), suggest a lengthening of the ice-free season along with a reduction of the sea ice extent during the Late Holocene. This is in agreement with other diatom records from the central RS[25] and the Antarctic Atlantic Ocean[43].

The contrasted patterns in WRS sea ice dynamics over the last millennia are mirrored in the TALDICE and TY δ[18]O (site temperature proxy) records, despite the relative proximity (550 km) of these two coastal core sites. TY δ[18]O record indicates a cooling at the ice core site throughout the Holocene, while TALDICE δ[18]O records suggest a warming since 8 ka (Fig. 6g). Here, we propose that the contrasted behaviours of the δ[18]O records in the late Holocene (from 7 to 8 ka onwards) result from different source areas and air mass pathways as observed today. Indeed, modern air mass trajectory analysis demonstrate that TALDICE precipitations originate from the Southern West Pacific–East Indian (including Northwestern RS), while TY precipitation is strongly influenced by the RS and TNB polynyas in Southwestern RS (Fig. 4). As such, the enhanced *d* and δ[18]O variability observed at TY during the past 7 ka (Fig. 3a) could be related to highly variable efficiency of the regional latent-heat polynyas, thus affecting storms and moisture origin. The positive correlation (*r* = 0.77) between TALDICE *d* and *F. curta* in WRS_JB (Fig. 6f) suggests that the moisture transport towards Talos Dome is affected by the sea ice extent in the OSIZ area, with higher *d* values related to a higher sea ice extent and main moisture sources located at lower latitudes.

These Holocene changes in pack ice cover, polynya and coastal sea ice variability are expected to have affected the regional polar marine ecosystems and oceanic primary production, thus triggering cascading effects on krill, fish, penguins and marine mammals[7]. Colony occupation of elephant seal (*Mirounga leonina*) and Adélie penguin (*Pygoscelis adeliae*), as well as the dietary contribution of Antarctic silverfish (*Pleuragramma antarcticum*) are documented with more than a hundred [14]C-dated samples around VLC[20, 21, 44] (Fig. 6a; Supplementary Fig. 3). The reproduction of seals and penguins are known to be highly sensitive to the distance between the nursery sites and the limit of the coastal sea ice[7, 45]. While penguins need pack ice for breeding success, elephant seals prefer open water close to nursery area. The presence of southern elephant seal colonies is interpreted to have been induced by significant decrease in coastal sea ice[20]. Successful foraging during chick rearing of Adélie penguins in WRS are linked to the abundance and distance of silverfish foraging areas[45] that are positively linked to pack ice/platelet ice.

Our new WRS_JB diatom data suggest that the RIS retreated between 11.8 and 9 ka with a complete opening of WRS by ~7–8 ka, in agreement with previous studies[40]. The opening of WRS allows the TNB colonies to be occupied by Adélie penguins by ~8.2–7.7 ka and elephant seals by ~7 ka. The penguin *Optimum* culmination (4.5–3.5 ka, Fig. 6a) and the concomitant increase in Antarctic silverfish[21] probably results from consolidated pack ice conditions as evidenced by highest *F. curta* relative abundances in WRS_JB core and elevated ssNa concentrations in TALDICE ice core. Conversely, the brief Seal *Optimum*[20] (2.5–1.1 ka) is synchronous to a warm event recorded both in the atmosphere (high

$\delta^{18}$O values in TALDICE and TY) and in the ocean/sea ice (multi-centennial reduction in TY ssNa and *F. curta* records as well as an increase in TRS occurrences) demonstrating less formation of new sea ice in latent-heat polynyas leading to less platelet ice and thinner coastal sea ice.

More persistent coastal sea ice, documented by high *F. curta* relative abundances in CSIZ cores and by high ssNa content in TY during the last millennium, is very likely responsible for the migration of Adélie penguin colonies from the Scott Coast to other sites in southern RS (Prior Island, South Bay and Northern Foothills) at 2.8 ka and the disappearance of elephant seal colonies in RS at ~1–0.5 ka[20] (Supplementary Fig. 3). We attribute these recent ecological variations to increased persistence of coastal sea ice due to enhanced efficiency of WRS latent-heat polynyas.

A coherent picture between marine and ice core records depicts contrasting changes of open vs. coastal sea ice during the past 7–6 ka in the WRS with a strong centennial to multi-centennial variability. It provides a coherent explanation for water isotopes trends in WRS ice cores and major Holocene faunal changes in, elephant seals, Adélie penguins and silverfish, which are most sensitive to sea ice conditions. Here, for the first time, we provide physical explanations for reconciling late Holocene marine, ice cores and faunal records in the WRS. Our observations point out that diatoms and faunal changes are directly correlated. The abundance/presence of TRS and elephant seals are related to the same favourable environmental conditions, as is the case for *F. curta* with Adélie penguins and silverfish. We also provide a physical explanation for differences between nearby Antarctic ice core records that are due to the specificity of air mass trajectories above different sea ice areas. The mechanisms that control changes in sea ice production and duration in this area imply an important role of katabatic winds in the efficiency of latent-heat polynyas. Recently, a circumpolar mapping of Antarctic coastal polynyas[11] confirms the relationship and feedback between coastal sea ice, katabatic winds and latent-heat polynya formation. Furthermore, it is demonstrated that a drastic change in coastal sea ice concentration and duration, which is particularly vulnerable to climate change, is linked to changes in the polynyas' efficiency and has possible implications on AABW formation and therefore on global climate.

Altogether, these results point out that katabatic winds are the major players in pack ice formation and thickening of landfast ice, while geostrophic winds are driving the export and extension of pack ice in WRS. This calls for the development of coupled ocean-atmosphere models that can incorporate the interplay between katabatic/geostrophic winds and sea ice formation in latent-heat polynyas and their export to the open ocean. Implementing the representation of the efficiency of sea ice factories in ocean-atmosphere models is critical for quantitative benchmarking of sea ice and AABW formation in WRS.

## Methods

**Location and description of the marine cores.** WRS_JB core (73°49′ S; 175°39′ E, 553 cm long) was retrieved at a water depth of 598 m in the Joides Basin, an elongated trough NE–SW oriented and separated into two parts by a sill located at about 75°S latitude[47]. In this paper, we consider the Holocene interval (above 244 cm), corresponding to 1.0–11.8 ka characterised by massive diatomaceous mud with sparse ice-rafted debris clasts. WRS_JB core is generally characterised by low magnetic susceptibility (MS) values (Fig. 7) (mean $85 \times 10^{-6}$ SI units). Higher values (from 100 to $400 \times 10^{-6}$ SI units) correspond to higher sand and gravel content, which are also evident in the X-ray radiographs (not shown). The organic carbon ($C_{org}$; Fig. 7) content varies from 0.6 to 1.4% with most values falling around 1.2%. Lower $C_{org}$ values were found in the sandy intervals. The molar ratio $C_{org}/N_{tot}$ (Fig. 7) varies from 3.6 to 12.2. It gradually increases from the bottom to the top of the core, reaching its highest values in the top 45 cm. The absolute diatom abundance (ADA; Fig. 7) shows a nearly homogenous distribution varying between 100 and $150 \times 10^6$ valves/gram dry weight (v/gdw). Relative abundances of

reworked diatom species are low with values between 0 and 1.5% of the total diatom assemblage including CRS.

WRS_CH core (72°18′ S, 170°03′ E, 443 cm long) was retrieved at a water depth of 456 m in the outer section of Edisto Inlet, a narrow trough NNE-SSW elongated with a maximum water depth of 500 m, in Cape Hallett area. The core is laminated throughout presenting silt to sandy-silt laminae that are alternated with fluffy diatomaceous mud. Sparse gravel fraction and some carbonate shells are present.

WRS_CH core is characterised by MS values (Fig. 7) ranging from 460 to $2330 \times 10^{-6}$ SI and exhibiting high variability above 140 cm. Excursions to higher values are due to layers/laminae richer in siliciclastic/volcanoclastic debris. The $C_{org}$ content (Fig. 7) shows a very homogenous distribution, varying from 0.3 to 0.7%. The $C_{org}/N_{tot}$ ratio similarly shows a very homogenous distribution varying from 4.1 to 9.0, with a mean value of about 7.0. No trend is discernible in the ADA record with most values ranging between 200 and $300 \times 10^6$ v/gdw. Reworked diatoms are almost absent.

WRS_WB core (74°11′ S, 166°03′ E, 463 cm long) was retrieved at a water depth of 1033 m in the inner part of Wood Bay. Morpho-bathymetric data highlight that sea floor morphology is characterised there by a narrow basin, deeper than 1000 m, oriented WNW-ESE and transversally connected by an 800 m deep sill to the Drygalski Basin. In this paper, we consider the hydrated laminated diatomaceous ooze section (0–409 cm), mainly black in colour alternated with lighter laminae having a cotton-like texture.

WRS_WB core is characterised by low MS values (mean $50 \times 10^{-6}$ SI units). The MS values (Fig. 7) gradually increase from 335 to 407 cm (100–$200 \times 10^{-6}$ SI units) in correspondence with the increase of terrigenous content. The $C_{org}$ content (Fig. 7) varies from 1 to 2% (mean 1.5%), except at the base of the core where $C_{org}$ values drop persistently below 1.5%. The $C_{org}/N_{tot}$ ratio record oscillates around 7.5 with rare peak values above 9.0 below 250 cm. The ADA varies from 297 to $2782 \times 10^6$ v/gdw with higher values recorded above 250 cm. No reworked diatom species were observed except in one level (97 cm).

**Chronology.** The chronology of WRS-CH and WRS-WB marine cores is based on 5 and 6, respectively, accelerator mass spectrometry (AMS) radiocarbon dates measured at the NOSAMS Laboratory of the Woods Hole Oceanographic Institution (Massachusetts, USA). For WRS_JB core, four previously published AMS dates have been used[47]. Radiocarbon dates were measured on acid insoluble organic matter (AIOM) from bulk sediment samples, except for one level in WRS_CH core, where a mollusc shell was present (Supplementary Table 1). The down-core dates were corrected by subtracting the core top ages, which embed the regional marine reservoir effect (MRE) and the local DC contamination, assuming that both MRE and DC did not change over the Holocene[46]. Since our cores were recovered using a gravity corer, which did not well preserve the water-sediment interface, we used the uncorrected ages of the superficial sediments of twin box cores (BAY05-bc21 in Cape Hallett and BAY05-bc40 in Wood Bay) to calculate the modern DC fraction at our core sites. For WRS_JB core lacking a twin box core, a superficial AIOM age of $3781 \pm 51$ yr, calculated by ref. [48], was used to correct the radiocarbon ages. The core top ages and $\delta^{13}$C value obtained in this study agree with the spatial distribution proposed by refs. [48,49] for the RS. The core top age for WRS_JB is bracketed by surface sediment ages in Joides Basin[47] with a standard deviation of $\pm608$ years. The core top age for WRS_WB shows a 200-year younger age than previously reported[48] for the WRS. However, the more northern and coastal location of WRS_WB compare to previous studies[48, 49] can explain this difference, whereby the local DC contamination is lessened by higher coastal productivity and a South–North decreasing gradient in DC contamination[48, 49]. The core top age obtained at WRS_CH agrees with those reported by ref. [50] and confirms the decreasing trend of DC contamination towards the northern WRS[48]. The corrected AMS $^{14}$C dates were then converted into calibrated ages using the CALIB REV 7.1.0 calibration program[51] and the MARINE13 calibration curve. The conventional, corrected and calibrated AMS $^{14}$C data are reported in Supplementary Table 1. The ages between dated levels were interpolated, assuming a linear sedimentation rate.

The arguments that support the $^{14}$C chronology are reported in the Supplementary Note 1.

Cores WRS_JB, WRS_CH and WRS_WB cover the last 11.8, 4.6 and 5.8 ka, respectively, with a mean sedimentation rate between 0.17 and 0.42 mm$a^{-1}$ for WRS_JB, between 0.64 and 2.87 mm$a^{-1}$ for WRS_CH and between 0.16 and 3.03 mm$a^{-1}$ for WRS_WB (Fig. 7).

**Diatoms.** A total of 264 samples were taken along the three marine cores for diatom analyses. Core WRS_JB was sampled every 2–4 cm from the top to the depth of 244 cm (1.0–11.8 ka), yielding a mean temporal resolution of ~150 years. Core WRS_CH was sampled every 4 cm with a mean temporal resolutions of ~20 years from 1 to 240 cm (0.1–1.4 ka) and ~60 years from 244 to 444 cm (1.5–4.6 ka). The samples were collected at intervals of 4–8 cm in core WRS_WB resulting in mean temporal resolutions of ~30 years from 2 to 301 cm (0–1.2 ka) and ~120 years from 305 to 409 cm (1.4–5.8 ka).

Sample preparation for diatom analyses followed the technique described in ref. [52]. At least 300 diatom valves in each sample were identified and counted at ×1000 magnification using an Ortholux light microscope with an oil immersion objective lens following the method outlined in ref. [53]. The most abundant and ecologically representative diatom species are shown in Fig. 2.

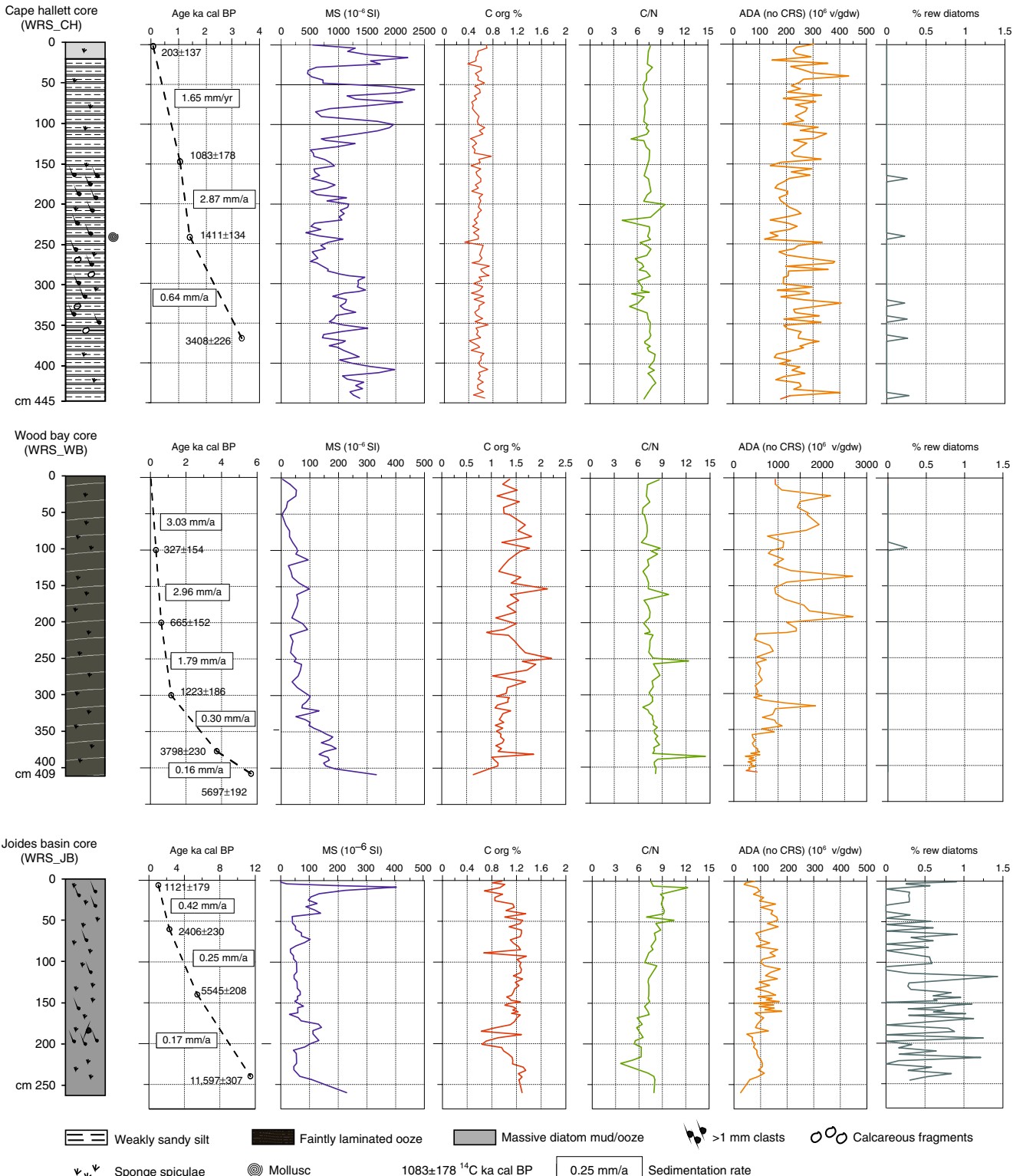

**Fig. 7** Marine core lithology and chronology. From top to bottom: description of cores WRS_CH, WRS_WB and WRS_JB showing from left to right, lithology, ages of radiocarbon-dated levels and sedimentation rates, magnetic susceptibility (MS), organic C content (C$_{org}$ %), organic C/total N molar ratio (C/N), absolute diatom abundance (ADA) expressed in number of valves per gram of dry weight (v/gdw), and percentage of reworked diatoms (% rew diatoms). The ADA has been calculated excluding the *Chaetoceros* resting spore (CRS)

**Ice core location and chronology**. A 1620 m deep ice core was drilled at Talos Dome in the framework of the Talos Dome Ice Core (TALDICE) project (www. taldice.org). The core provides a palaeoclimate record covering the past 250 ka back to marine isotope stage (MIS) 7.5[36]. Here we consider the uppermost section between 4 and 687 m corresponding roughly to the Holocene period.

Talos Dome is a peripheral dome of East Antarctica, located in the RS sector about 250 km from the Southern Ocean, 550 km North of TY and 1100 km East of Dome C, and it is characterised by a snow accumulation rate of 80 mm water equivalent per year (average 2004–1259 AD)[36], about three to four times higher than East Antarctic plateau sites. For the TALDICE ice core we have adopted here the age scale proposed for the Holocene in ref. [54] using volcanic matching.

TY ice core records (http://isolab.ess.washington.edu/isolab/taylor/) cover the past ~130 ka back to MIS 5e, and is located ~150 km from the western edge of RS, inland of the Transantarctic Mountains. Today, it is characterised by a snow accumulation rate of 65 mm water equivalent per year[34].

Since the official age scale st9810[55], adopted for the TY ice core, has been recently updated[56] for the last 2 ka using a sulphate volcanic synchronization method, we merged two chronologies: for the last 1.7 ka the chronology, as described in ref. [56], is based on volcanic synchronization, while for the older part, we used the chronology as described in ref. [57], which is based on gas synchronization. In the overlapping period, the two chronologies present an age difference of only 10 years.

**Ice core data.** The TALDICE ice core sea salt sodium (ssNa) and water stable isotope records is obtained along 1 m resolution samples ('bag samples'), which correspond to a mean Holocene temporal sampling resolution of 18 years, ranging between 11 years over the last 2 ka and 24 years over the early Holocene (9–12 ka).

Measurements of major ions were performed by means of ion chromatography, with an overall accuracy of the method can be estimated to be around 5%[58]. Na content in Antarctic aerosol is always dominated by sea salt, especially in interglacial periods, showing a minor contribution of dust leachable Na. ssNa has been calculated using non sea salt Ca (nssCa) as crustal marker and a simple two-variable, two-equation system allowing evaluating the ss- and nss-fractions of Na and Ca:

$$totNa = ssNa + nssNa \qquad (1)$$

$$totCa = ssCa + nssCa \qquad (2)$$

$$ssNa = totNa - 0.562\,nssCa \qquad (3)$$

$$nssCa = totCa - 0.038\,ssNa, \qquad (4)$$

where 0.562 = Na/Ca (w/w) in the crust and 0.038 = Ca/Na (w/w) in seawater.

The raw data of the ssNa content obtained from bag samples of the TALDICE ice core for the period corresponding to the Holocene are reported in the Supplementary Fig. 4.

Measurements of the two water stable isotopes ($\delta$D and $\delta^{18}$O) were conducted on continuous bag samples. The TALDICE $\delta^{18}$O (published in refs. [36,41]) and new $\delta$D measurements were conducted using the well-known $CO_2$ ($H_2$)/water equilibration technique. The accuracy of $\delta^{18}$O and $\delta$D measurements is $\pm0.05$‰ ($1\sigma$) and $\pm0.7$‰ ($1\sigma$), respectively, leading to a final precision of $\pm0.8$ ‰ on deuterium excess ($d = \delta$D$-8 \times \delta^{18}$O). The raw data of the deuterium excess obtained from the bag samples of the TALDICE ice core for the period corresponding to the Holocene are reported in the Supplementary Fig. 5.

**Back trajectory analysis.** A climatology of 5 days backward trajectories ending at 1000 m above ice core site of TALDICE and TY was generated using the Hybrid Single-Particle Lagrangian Integrated Trajectory (HYSPLIT4) model[59] for the period 1979–2012. The ERA-Interim reanalysis[60] archived as meteorological gridded data set was used to initialise the HYSPLIT4 model. The ERA-Interim reanalysis products are provided by the European Centre for Medium-range Weather Forecasts (ECMWF) and they are available every 6 h with 1° latitude–longitude resolution[60]. We computed one trajectory per day (ending at 12 UTC above the site). Due to the uncertainty associated to trajectories longer than 5 days, which is estimated between 10 and 30% of the travel distance[32], the analysis was limited to the first 3 days of the trajectories with the uncertainty reduced significantly. ERA-Interim reanalysis is one of the most reliable atmospheric data sets covering the entire Antarctic Ocean and the last 40 years since the satellite era (1979–present). The most intense katabatic winds, that form the latent-heat polynyas, are influenced by small-scale orography such as coastal valleys. During the last 5 ka, the period analysed in detail in this paper, the geographic configuration of the Western RS and the adjacent ice sheet (ice shelf and glacier valleys position, ice sheet elevation, etc.) were very similar to the modern situation and we could assume similar patterns for latent-heat polynya positions, katabatic wind confluence zones and, consequently, for the main ssNa source areas for each ice core. However, changes in polynya efficiency/katabatic wind feedback mechanisms, sea ice cover/extension and geostrophic wind could have changed the percentage of ssNa from source areas at each ice core site. On the basis of the back trajectory analysis, it was observed that more of the 40% of ssNa at TY and <30% at TALDICE comes from latent-heat polynyas, but it must be considered that any changes in the polynya efficiency during the past could reduce significantly the ssNa coming from the polynyas, in particular for the TY site.

Each back trajectory was projected on the Sea Ice Concentration field (SIC) and on the 10-m wind field (still provided by the ECMWF-ERA-Interim reanalysis), associating each value along the trajectory path with the nearest SIC and wind speed values[61]. In order to identify air masses presumably responsible for the sea ice-related ssNa transport towards the ice core sites, a selection was carried out taking into account: (i) trajectories which spent at least 10% of their path over the

sea and <40% over the continent; (ii) trajectories computed for the period corresponding to the beginning of the sea ice formation, its growth until its maximum extent (March–November); (iii) trajectories which spent at least 3 h consecutively above sea ice, diagnosed for grid cells where SIC fractions are >15%.

This selection is refined to calculate the density of back trajectories, which are considered as loaded by ssNa above the sea ice. Loading conditions are defined if the sea ice fraction ranges between 15 and 75 % and the 10-m wind speed >3 m s$^{-1}$. The density of events is spatially cumulated for a regular grid of 5° × 1° of longitude and latitude.

**EOF analysis.** EOF analysis reduces the dimensionality of a data set consisting of a large number of interrelated variables retaining as much as possible of the variation present in a data set. This is achieved by transforming data into a new set of variables, the principal components (EOFs), which are orthogonal to each other, and ordered so that the first few EOFs retain most of the variation present in all the original variables[62].

As a first step, the original data matrix F is first compressed into a covariance matrix (R), and then the eigenvalue problem is solved as shown below:

$$R\,EOF = \lambda\,EOF. \qquad (5)$$

The data matrix F can accordingly be decoupled in:

$$F = \sum_j PC_j \times EOF_j, \qquad (6)$$

where $\lambda$ is a diagonal matrix containing the eigenvalues. $\lambda_j$ and $EOF_j$ represents corresponding eigenvectors, $PC_j$ represents principal component time series and highlights the time evolution of the corresponding $EOF_j$. For a detailed description of mathematical fundamental of EOF analysis see ref. [63].

The time series of $\delta^{18}$O, deuterium excess and ssNa obtained from ice cores and *F. curta* obtained from marine cores, are used as input data matrix for the EOF analysis, after a 200-year resampling and a standardisation to z-scores. Least squares empirical orthogonal functions approach (LSEOF) is applied[64] to decompose the data matrix, which contains missing values due to different lengths of the input records. Several issues were identified with the use of this approach[65]. A covariance matrix derived from data with different time coverage is not necessarily positive defined and the decomposition via LSEOF can contain negative values of $\lambda$ leading to a subsequent overestimation of EOFs amplitude and of the amount of explained variance contained therein[66]. Furthermore, the decomposition of a non-positive defined covariance matrix can lead to a loss of orthogonality between EOFs[66]. Figure 5 shows the first two EOF spatial modes and the explained variance obtained with LSEOF analysis. The calculated eigenvalues lead to non-negative values and identify two modes, well separated from the others, and explaining more than 70% of the total variance (43% and 28% for EOF1 and EOF2, respectively). These results point out the absence of an overestimation in the EOF1 and EOF2 analysis and reveal a no strong degeneracy between different modes. EOF1 highlights a high negative value for TY $\delta^{18}$O and high positive value for TALDICE and TY deuterium excess proxies, TALDICE $\delta^{18}$O and ssNa, and WRS_CH, WRS_WB, WRS_JB proxies (*F. curta*). Conversely EOF2 represents a superimposed signal with a high negative score for WRS_CH, WRS_WB proxies (*F. curta*) and TY ssNa, and high positive WRS_JB proxy (*F. curta*) and the other records characterised by slight positive values.

$PC_1$ and $PC_2$ time series also reveal different trends (not shown). However, PC time series are less reliable than EOF spatial pattern due to the limited number of overlapping records in the EOFs analysis.

**Correlation analysis.** All the data have been resampled at a common 200-year step using the AnalySeries software[67] and standardized ((x-mean)/stdev) for the considered period. The linear correlation tables (Supplementary Tables 3–7) have been produced using PAST—paleontological statistics, version 3.01[68].

**Data availability.** The raw data generated during the current study are available in the NOAA World Data Center for Paleoclimatology (WDC Paleo) at the following link: https://www.ncdc.noaa.gov/paleo/study/22590

The data sets for the TY ice core are available at the Taylor Dome Ice Core Project web site, link for chemistry data: http://isolab.ess.washington.edu/isolab/taylor/data/tdchem.dat; link for water stable isotope data: http://isolab.ess.washington.edu/isolab/taylor/data/tdxs.dat.

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

## Acknowledgements

This research was conducted in the framework of the PNRA-MIUR funding scheme and contributed to the European Science Foundation HOLOCLIP Project and TALDICE Project. The HOLOCLIP Project, a joint research project of the ESF PolarCLIMATE programme, is funded by national contributions from Italy, France, Germany, Spain, Netherlands, Belgium and the United Kingdom. TALDICE, a joint European programme led by Italy, is funded by national contributions from Italy, France, Germany, Switzerland and the United Kingdom. This is HOLOCLIP publication 30 and TALDICE publication 48.

## Author contributions

K.M., X.C., E.C. and R.M. contributed to the new diatom measurements and marine core dating. B.S., M.B., V.M.-D., M.S. and R.T. contributed to the new TALDICE deuterium excess and ssNa measurements. C.S., V.C. and M.F. contributed to the atmospheric modelling. C.B. and M.C.S. provided faunal remains distributions and dating. K.M., B.S., M.F., X.C. and C.B. worked on the comparisons of different records and their interpretations, wrote most of the manuscript and prepared the figures with significant contributions from V.M.-D., V.C., C.S., E.C. and R.M.

## Additional information

**Competing interests:** The authors declare no competing financial interests.

