## [Peer Review File · Nature Communications]

Reviewers' comments:

Reviewer #1 (Remarks to the Author):

I appreciate the chance to take a third look at the revised version of the manuscript, "Holocene wind and sea ice variability in Western Ross Sea (Antarctica)." The authors have addressed the concerns that I detailed previously, most critically through their discussion of the potential problems with the marine sediment core chronologies, with the addition of both text material and figure 7. I understand the position the authors are in, that of course, more could be done to potentially improve/re-assess the chronologies with advanced technologies such as ramped pyrolysis - but that at some point we all need to move forward with what we have. The authors make a solid case and demonstrate that their interpretations are sound even with uncertainties in dating. The figures are improved with regard to clarity, and the scope of the paper, with a focus on the Ross Sea, is appropriate. In terms of broad scale interest, this study is an excellent example of working with marine and terrestrially-based paleoclimate records to understand regional history, as well as of interest in terms of its application to the larger ecosystem, through the macrofaunal records. I have no additional suggestions for changes except for a few very minor mistakes (spelling and grammatical) that are present, which will be caught by a copy editor.

Reviewer #2 (Remarks to the Author):

Mezgec and co-authors compare sediment core records (proxies for sea ice), faunal remains (dates of colonization by seals and penguins), and ice core records (climate proxies) to ascertain the role of past sea ice changes on regional ecology in the Ross Sea, Antarctica. Linking these various records is important because, as the authors state, sea ice presence and polynya efficiency carry importance for global climate feedbacks. Implementation of sea ice factories efficiency in ocean-atmosphere models is a main recommendation of the paper.

I found the use of these three types of records in tandem very interesting, however it is very difficult to link them together. The primary issue is the chronology of the sediment cores. The authors point out that no further analyses are possible (indeed they have done a lot of work), but that there are better dating methods now available. They discuss their dates in the response to reviewers, and ascertain that the cores are reliable to about 650 y precision on the dates (slightly higher than what is currently listed in the methods section of the manuscript, 200-500y). But central in their conclusion about the ages is an assumption that Dead Carbon (DC, carbon that is pre-aged, or even free of ^{14}C) is constant downcore. If there is any region where this may be true, it is the Ross Sea with very high productivity and larger distances to the continent. However, previous work from the Antarctic Peninsula shows that, even when lithology and basic geochemistry (C:N, $\delta^{13}\text{C}$, etc) are constant, the DC contribution can be quite variable. This could still impart unaccounted for uncertainty to the sediment core ages, making them larger. The authors and editors have to wonder - does this story change if the ages of the sediment cores are more uncertain than depicted in the manuscript? If uncertainty in the ages increases, at what point do the authors' conclusions become questionable? Does it happen before or after a threshold of likely uncertainty is crossed? This may be the last sort of exercise or calculation that the authors' could do in lieu of more age measurements.

A more minor critique of this manuscript is the use of Lagrangian modeling. It is, on the surface, an approach that seems to be on the cutting edge. However, it is a model. It may be useful, but shouldn't be deemed correct. The authors use this model to suggest the main pathways of air traveling over the sites given modern (since 1979) sea ice and satellite observations. A more interesting use of this model would be to feed it different hypothetical sea ice geometries as suggested in lines 169-172 to see if there is a significant feedback to wind patterns. In doing so,

the authors would constrain whether their approach is good or not, and they would address one of their most important points (that sea-ice factories efficiency should be fed into coupled climate change models). In doing so, though, there would be a risk that they were correct in using modern wind trajectories to ascertain past patterns during sea-ice shifts, but they would have to argue against the importance of improving parameterization of such sea ice efficiency feedbacks into models.

I appreciated this manuscript despite the flaws in the marine sediment core chronology and the use of the air trajectory model. I think more effort should go into linearly explaining each of the three types of records and the use of the model to make the main message more poignant.

Otherwise, the average reader (as well as the expert) needs to skip around between the various sections in the currently-formatted manuscript to make sense of the authors' conclusions and discussion. This is not desirable. A case in point is the well-summarized response to reviewers.

This should be part of the main manuscript. It makes some pertinent arguments, but also spells out the issues with the chronology. Without this in the published article, those arguments are lost from the readership and the paper becomes less important (or even controversial).

In sum, I don't think that compound specific or Ramped PyrOx will solve all of the authors' issues, but it certainly may clear some issues up. Even just another bulk age from the same core depth as the mollusc shell would have helped test their core-top assumptions. It is a fascinating study, but Nature Communications may not be the right venue to fully discuss all of the intricacies in linking these records.

The reviewer comments are in red, while our answers are in black.

Reviewer #1 (Remarks to the Author):

I appreciate the chance to take a third look at the revised version of the manuscript, "Holocene wind and sea ice variability in Western Ross Sea (Antarctica)." The authors have addressed the concerns that I detailed previously, most critically through their discussion of the potential problems with the marine sediment core chronologies, with the addition of both text material and figure 7. I understand the position the authors are in, that of course, more could be done to potentially improve/re-assess the chronologies with advanced technologies such as ramped pyrolysis - but that at some point we all need to move forward with what we have. The authors make a solid case and demonstrate that their interpretations are sound even with uncertainties in dating.

We thank the reviewer for his/her time spent on our paper that was greatly improved for his/her comments. We implement the chronology section in the methods and explain in the Supplementary Information Note 1 all the caveats with ^{14}C chronology in the Ross Sea, and a sensitivity test.

The figures are improved with regard to clarity, and the scope of the paper, with a focus on the Ross Sea, is appropriate. In terms of broad scale interest, this study is an excellent example of working with marine and terrestrially-based paleoclimate records to understand regional history, as well as of interest in terms of its application to the larger ecosystem, through the macrofaunal records. I have no additional suggestions for changes except for a few very minor mistakes (spelling and grammatical) that are present, which will be caught by a copy editor.

We thank reviewer 1 for his/her appreciations. We nonetheless further improved the clarity of the figures and rearranged them according to the new structure of the paper requested by reviewer 2.

Reviewer #2 (Remarks to the Author):

Mezgec and co-authors compare sediment core records (proxies for sea ice), faunal remains (dates of colonization by seals and penguins), and ice core records (climate proxies) to ascertain the role of past sea ice changes on regional ecology in the Ross Sea, Antarctica. Linking these various records is important because, as the authors state, sea ice presence and polynya efficiency carry importance for global climate feedbacks. Implementation of sea ice factories efficiency in ocean-atmosphere models is a main recommendation of the paper.

I found the use of these three types of records in tandem very interesting, however it is very difficult to link them together. The primary issue is the chronology of the sediment cores. The authors point out that no further analyses are possible (indeed they have done a lot of work), but that there are better dating methods now available.

Although we are aware that now better methods are available, we cannot perform further analyses. To note though that even these new dating techniques present their own flaws. In the Supplementary Information, we added a section detailing how robust are the marine core chronologies along with some arguments on the benefit and flaws of new dating techniques. Indeed, it was observed that ^{14}C of fatty acids can also be altered by old carbon as dates as old as 22 ka BP

have been measured in the laminated sections above the diamicton at site U1357 off Adélie Land (Ohkouchi, personal communication).

They discuss their dates in the response to reviewers, and ascertain that the cores are reliable to about 650 y precision on the dates (slightly higher than what is currently listed in the methods section of the manuscript, 200-500y).

We better explained all the caveats related to ^{14}C dating and uncertainties in the Ross Sea moving what reported in the previous response letter to the Chronology section in Methods and in the Note 1 of Supplementary Information.

But central in their conclusion about the ages is an assumption that Dead Carbon (DC, carbon that is pre-aged, or even free of ^{14}C) is constant downcore. If there is any region where this may be true, it is the Ross Sea with very high productivity and larger distances to the continent. However, previous work from the Antarctic Peninsula shows that, even when lithology and basic geochemistry (C:N, $\delta^{13}\text{C}$, etc) are constant, the DC contribution can be quite variable. This could still impart unaccounted for uncertainty to the sediment core ages, making them larger.

As reviewer 2 stated, very high productivity in Ross Sea may reduce DC variations. Although limited in numbers, previous studies suggest a rather constant offset between AIOM and CaCO_3 ages (~400 years) in the Ross Sea (Andrews et al., 1999) and the similarly productive Adélie Land region (Costa et al., 2007; Dunbar, personal communication). The Antarctic Peninsula is a very different environment with higher ice free areas (Lee et al., 2017) and much more contorted coasts that indeed increase the possibility of variable DC to the marine environment in response to deglaciation. Although Ross Sea deglaciation may have impacted early Holocene ages at JB site, the ice sheet was well southward at 7 ka BP when OSIZ and CSIZ already diverged. Additionally, even a couple of hundred years of DC changes will not change our interpretations as proved below by the sensitivity tests (see next paragraph).

The authors and editors have to wonder - does this story change if the ages of the sediment cores are more uncertain than depicted in the manuscript? If uncertainty in the ages increases, at what point do the authors' conclusions become questionable? Does it happen before or after a threshold of likely uncertainty is crossed? This may be the last sort of exercise or calculation that the authors' could do in lieu of more age measurements.

We add in the supplementary information the figure, reported in the previous response letter, showing our sensitivity test adding ± 600 year to the marine records and adding ± 100 years to the ice core records, supporting our main results even if the cores are moved in opposite temporal direction.

A lag correlation analysis where *F. curta* records are moved forward or backward by 200 years, 400 years and 600 years compare to TY and TD ssNa records further demonstrates that our interpretations are still valid at this "inaccuracy".

A more minor critique of this manuscript is the use of Lagrangian modeling. It is, on the surface, an approach that seems to be on the cutting edge. However, it is a model. It may be useful, but

shouldn't be deemed correct. The authors use this model to suggest the main pathways of air traveling over the sites given modern (since 1979) sea ice and satellite observations. A more interesting use of this model would be to feed it different hypothetical sea ice geometries as suggested in lines 169-172 to see if there is a significant feedback to wind patterns. In doing so, the authors would constrain whether their approach is good or not, and they would address one of their most important points (that sea-ice factories efficiency should be fed into coupled climate change models). In doing so, though, there would be a risk that they were correct in using modern wind trajectories to ascertain past patterns during sea-ice shifts, but they would have to argue against the importance of improving parameterization of such sea ice efficiency feedbacks into models.

The text in the *methods* section has been improved to constrain the quality of the approach, whereas it is not possible to implement the parameterisation proposed by the reviewer in the "model" employed for our analysis.

The *Lagrangian modeling* used in the manuscript is the HYSPLIT 4 model initialised by meteorological gridded dataset ERA-INTERIM. HYSPLIT is a complete system for computing simple air parcel trajectories, as well as complex transport, dispersion, chemical transformation, and deposition simulations. HYSPLIT is one of the most extensively used atmospheric transport and dispersion models in the atmospheric sciences community. ERA-Interim is a climate reanalysis gives a numerical description of the recent climate (for Antarctica since 1979), produced by combining models with observations using data assimilation system. It contains estimates of atmospheric parameters (temp, air press., wind, etc.) and sea ice observation coming by satellite observations (e.g. microwave). The sea ice extension is not parameterised but it is provided (as the concentration) from satellite image analysis. Therefore, our analysis of back-trajectories, using the ERA-Interim, does not permit to parameterised the sea ice extension/concentration and related feedback with meteorological conditions (e.g. wind etc.) because these values come from the assimilation in the model of the satellite data. Aiming to answer to the request of Reviewer concerning the possibility to change the polynya geometries into the model, it should be used coupled climate change models (e.g. LOVECLIM, IPSL-CMIP5 etc.); however, these models present a spatial resolution two or three times greater ($3^{\circ} \times 3^{\circ}$ in lat/ lon i.e. thousands of km) than ERA-INTERIM reanalysis ($1^{\circ} \times 1^{\circ}$). Actually CGCMs are not able to spatially resolve the polynya areas (hundreds of km) and also the glacier valleys (tens of km) that channelize the katabatic wind. Lastly, several authors have already pointed out the straight correlation between katabatic wind and polynya "efficiency" on sea ice and HSSW productions (Zwally et al. 1985; Pease 1987; Adolphs and Wendler 1995; Markus and Burns 1995; Massom et al. 1998; Comiso et al. 2011; Drucker et al. 2011). Moreover, Gallée (1997) using a model, pointed out a strong positive feedback between the katabatic wind system and the latent heat flux polynya which reinforces the katabatic wind coming off the ice sheet. The improvement of these CGCMs to resolve the sea ice production/dynamics is beyond the scope of the manuscript.

I appreciated this manuscript despite the flaws in the marine sediment core chronology and the use of the air trajectory model. I think more effort should go into linearly explaining each of the three types of records and the use of the model to make the main message more poignant. Otherwise, the average reader (as well as the expert) needs to skip around between the various sections in the currently-formatted manuscript to make sense of the authors' conclusions and discussion. This is not desirable.

We significantly reorganized the results and discussion part to make the flow of the reading more linear.

A case in point is the well-summarized response to reviewers. This should be part of the main manuscript. It makes some pertinent arguments, but also spells out the issues with the chronology. Without this in the published article, those arguments are lost from the readership and the paper becomes less important (or even controversial).

We moved parts of the previous response letter to the main text, to the Chronology section, as well as to the Supplementary Information. We now believe that the readership will have all the necessary information in hands.

In sum, I don't think that compound specific or Ramped PyrOx will solve all of the authors' issues, but it certainly may clear some issues up. Even just another bulk age from the same core depth as the mollusc shell would have helped test their core-top assumptions. It is a fascinating study, but Nature Communications may not be the right venue to fully discuss all of the intricacies in linking these records.

We do our best in order to re-structure the manuscript in the Nature Communication style and we really think that now the reading of this new revised version is more linear and poignant, making the manuscript a fascinating study.